# Connecting the Dots between Schizotypal Symptoms and Social Anxiety in Youth with an Extra X Chromosome: A Mediating Role for Catastrophizing

**DOI:** 10.3390/brainsci7090113

**Published:** 2017-09-06

**Authors:** Anne C. Miers, Tim Ziermans, Sophie van Rijn

**Affiliations:** 1Developmental and Educational Psychology Unit, Institute of Psychology, Leiden University, P.O. Box 9555, 2300 RB Leiden, The Netherlands; 2Clinical Child and Adolescent Studies, Leiden University, P.O. Box 9555, 2300 RB Leiden, The Netherlands; t.b.ziermans@uva.nl (T.Z.); srijn@fsw.leidenuniv.nl (S.v.R.); 3Department of Psychology, Brain and Cognition, University of Amsterdam, Amsterdam, The Netherlands; 4Leiden Institute for Brain and Cognition (LIBC), Leiden University, P.O. Box 9600, 2300 RC Leiden, The Netherlands

**Keywords:** Klinefelter, Trisomy X, schizotypal symptoms, social anxiety symptoms, catastrophizing coping

## Abstract

Youth with an extra X chromosome (47, XXY & 47, XXX) display higher levels of schizotypal symptoms and social anxiety as compared to typically developing youth. It is likely that the extra X chromosome group is at-risk for clinical levels of schizotypy and social anxiety. Hence, this study investigated how schizotypal and social anxiety symptoms are related and mechanisms that may explain their association in a group of 38 children and adolescents with an extra X chromosome and a comparison group of 109 typically developing peers (8–19 years). Three cognitive coping strategies were investigated as potential mediators, rumination, catastrophizing, and other-blame. Moderated mediation analyses revealed that the relationship between schizotypal symptoms and social anxiety was mediated by catastrophizing coping in the extra X chromosome group but not in the comparison group. The results suggest that youth with an extra X chromosome with schizotypal symptoms could benefit from an intervention to weaken the tendency to catastrophize life events as a way of reducing the likelihood of social anxiety symptoms.

## 1. Introduction

This study reports on psychological processes that may explain the development of social anxiety in a group of children and adolescents at increased risk for elevated schizotypal symptoms: individuals with an extra X chromosome. The additional X chromosome is caused by non-disjunction during gametogenesis and leads to the 47, XXY karyotype in boys (Klinefelter syndrome) and the 47, XXX karyotype in girls (Trisomy X). There is an increasing interest in this genetic condition, fueled by accumulating evidence over the past decades that having an extra X chromosome not only impacts physical development, but also psychological development. Indeed, considering the exceptionally high density of genes on the X chromosome that are essential for the development of the brain [1], individuals with Klinefelter syndrome and Trisomy X are at risk for neurodevelopmental problems.

With regard to specific genetic X linked mechanisms that may explain the Klinefelter and Trisomy X phenotypes, it has been proposed that (i) parental origin of the X chromosome might play a role; (ii) X linked genes that escape inactivation may be involved, and (iii) these genes may lie in the pseudo-autosomal region (PAR) and hence have a homologue on the Y chromosome [2]. Which exact genes are overexpressed is still largely unknown, although recent research has revealed overexpression of NLGN4Y, a gene that may be involved in neural synaptic function [3] and increased copy number variations [4] in Klinefelter syndrome. Risk for psychopathology, including anxiety and schizotypal symptoms, in Klinefelter syndrome is thus thought to be anchored in altered development of the brain. This is also evident from structural and functional magnetic resonance imaging (fMRI) studies showing neural abnormalities, specifically in the temporal and frontal regions, consistent with the cognitive phenotype associated with Klinefelter syndrome that is characterized by impairments in language, executive functioning and social cognition [5,6,7,8,9].

Studies of children and adults with an extra X chromosome show that on average the level of schizotypal traits is significantly higher than is found in the general population [10,11]. Schizotypy represents a continuum of personality characteristics and experiences ranging from normal dissociative, imaginative states to more extreme states related to psychosis and, in particular, schizophrenia. Indeed, the risk for schizophrenia has been found to be somewhat higher (odds ratio 3.6) in a sample of 860 adults with XXY, as compared to the general population [12], and in a sample of 51 boys with XXY (aged 6 to 19 years), 12% were diagnosed with a psychotic disorder not otherwise specified (NOS) or Schizoaffective disorder based on psychiatric screening with the Kiddie-SADS [13]. In addition to increased risk for schizotypal symptoms, substantially increased levels of social anxiety were also found in this population with, as compared to a healthy control group, an effect size of d = 0.8 [14]. In the present study, we extended previous research on the social behavioral phenotype of youth (e.g., [14]) with an extra X chromosome by investigating how schizotypal and social anxiety symptoms are related in this group and emotion regulation mechanisms that may explain their association, to better identify specific risk mechanisms.

We chose to explore cognitive emotion regulation strategies as mediating mechanisms between schizotypal symptoms and social anxiety for three reasons: (i) experiencing schizotypal symptoms, particularly along the positive dimension, can be threatening and stressful, and emotion regulation strategies are employed in the context of threatening and stressful events [15]; (ii) emotion regulation strategies are proposed to be a vulnerability factor in the vulnerability–stress model of psychotic episodes [16]; and (iii) several studies have highlighted the importance of cognitive emotional regulation strategies in relation to internalizing psychopathology [15]. Emotion regulation refers to how individuals manage emotionally arousing information [17]. Cognitive strategies describe those that are internally driven, characterized by thoughts about oneself, one’s feelings and others [18]. In this study, we focused on strategies that are consciously accessible and as measured by the Cognitive Emotion Regulation Questionnaire for Children (CERQ-C; [18]). This questionnaire has nine strategies that are often categorized into adaptive (acceptance; positive refocusing; refocus on planning, positive reappraisal; and putting into perspective) and maladaptive (self-blame; rumination; catastrophizing; and blaming others). The term ‘cognitive emotion regulation’ can be used interchangeably with the term ‘cognitive coping’ [19], and the latter term is used hereafter. From the age of eight or nine years, children employ cognitive coping strategies to deal with life’s challenges and stressors [18].

Relevant to the present research are studies that focus on cognitive coping strategy use in youth at-risk for developing psychotic disorder. Lin et al. [20] examined longitudinal relationships between subclinical positive psychotic experiences and coping in a large adolescent population (*n* = 813). The participants were assessed three times, spanning a total of three years. Three coping styles were included, task-, emotion- and avoidance-oriented. The emotion-oriented styles describe self-oriented emotional reactions and self-preoccupation and fantasizing, similar to the CERQ cognitive strategies of self-blame and rumination [19,21]. Results showed that subclinical positive psychotic experiences predicted emotion-oriented coping over time [20]. Further, individuals classified as following a persistent subclinical psychotic experiences pathway reported both the highest mean levels of emotion-oriented coping and the highest proportional use of emotion-oriented coping. These results are supported by Jalbrzikowski and colleagues’ [22] study in which youth (*n* = 88) at risk for psychosis were compared to a group of healthy controls (*n* = 53) cross-sectionally and longitudinally over a 12-month period. In that study, coping strategies were categorized into maladaptive and adaptive, with the maladaptive strategies conceptually related to CERQ self-blame and rumination. The cross-sectional analyses showed that a greater use of maladaptive coping typified the at-risk group. Over time, significant associations were found between maladaptive coping and concurrent schizotypal symptoms in the at-risk group [22]. Overall, these studies are quite consistent in their findings that the cognitive strategies self-blame and rumination are positively associated with schizotypal symptoms.

In terms of internalizing psychopathology and cognitive coping, much attention has been paid to symptoms of depression and anxiety, in both non-clinical and clinical youth populations. In a first study examining cognitive coping strategies in 9–11-year-old children from the general population, Garnefski et al. [18] pointed to the importance of self-blame, catastrophizing, and rumination in relation to self-reported worry and fearfulness. Legerstee et al. [23] showed that the same three strategies accounted for most of the variance in a comparison between a group of adolescents diagnosed with an anxiety disorder compared to a control group; without a diagnosis Garnefski and Kraaij [15] examined the specificity of relations between cognitive coping strategies and anxiety symptoms in a sample of adolescents, aged 13 to 16 years. The results showed that two strategies were uniquely associated with anxiety symptoms, after controlling for depression: catastrophizing and blaming others. The finding that catastrophizing is uniquely related to anxiety thus confirms the results of the Garnefski et al. [18] study, whereas the unique relation with other-blame was not suggested in that study.

As regards the relationship between cognitive coping strategies and social anxiety specifically, Miers et al. [24] investigated whether rumination after a stressful speech task was related to an increase in avoidance of social situations during adolescence. Social avoidance behavior is a key criterion of the diagnosis of social anxiety disorder (SAD; [25]) and is potentially of crucial importance to explaining the adolescent-onset of SAD. Miers et al. [24] showed that rumination significantly discriminated between youth following an increased avoidance pathway and youth following a consistently low avoidance pathway. This means that youth who ruminated more frequently after the speech task were more likely to increasingly avoid social situations during the adolescent period. In addition, this study also reported a significant positive correlation between self-reported social anxiety and rumination [24].

In sum, research suggests that youth with an extra X chromosome are at an increased risk for developing psychopathology, both schizotypal symptoms and social anxiety. Because the literature to date has reported on these phenotype characteristics independently, this study will bring these two strands together in the following two aims: (i) to investigate the relationships between schizotypal symptoms, social anxiety, and cognitive coping styles in a group of youth with an extra X chromosome and a group of typically developing peers; and (ii) to explore mediators of the relationship between schizotypal symptoms and social anxiety: which coping styles explain this relationship, and is the mediation contingent on group? Based on research here reviewed, the coping styles rumination, catastrophizing and blaming others [15,20,24] are particularly relevant to schizotypal symptoms and (social) anxiety. We therefore chose to investigate these three strategies as possible mediators of the relation between schizotypal symptoms and social anxiety.

## 2. Materials and Methods

### 2.1. Participants and Procedure

The current study’s sample consisted of 38 youth with an extra X chromosome (22 boys with Klinefelter syndrome and 16 girls with Trisomy X) and 109 non-clinical controls (47 boys and 62 girls). The participants were 8–19 years old. Youth with an extra X chromosome were recruited through one of two ways: (i) active follow-up of families after prenatal diagnosis with the help of academic medical centers in the Netherlands and Belgium. These clinical genetic departments screened their databases for families who had received a prenatal diagnosis of Klinefelter syndrome or Trisomy X; (ii) families seeking help for developmental problems (referral by pediatricians, psychologists, psychiatrists and clinical genetics departments) or for information about the condition of their child (through support groups and calls for participants). Diagnosis if Klinefelter syndrome and Trisomy X was confirmed by standard karyotyping, all had non-mosaic karyotypes.

The group of non-clinical controls were recruited from schools located in the western part of the Netherlands. These youth were screened for psychopathology and none scored in the clinical range on the Child Behavior Checklist [26].

The participants in this study are drawn from a larger sample, details of which are reported in other studies [14,27]. Inclusion criteria for this study’s sample were Dutch as the primary language, absence of neurological conditions (including for example structural brain damage, traumatic head injury, tumors, stroke or infections, and neurological diseases affecting the central nervous system), age 8 years and older, and an IQ of ≥70. An IQ of ≥70 is 2 *SD*s below the mean, based on the normal intelligence distribution. We selected this cut-off in order to exclude participants for whom answering the questionnaires would be too cognitively challenging. Table 1 presents the characteristics of the two groups. The groups were similar in terms of their gender distribution, mean age, and level of parental education. The groups differed significantly on IQ, with the control group scoring higher on IQ than the extra X chromosome group.

After providing a complete description of the study to the participants and their parents, we obtained written informed consent according to the Declaration of Helsinki. All assessments were conducted in a quiet room at Leiden University or one of the Academic Medical Centers in the Netherlands. All data were obtained (under supervision of a trained psychologist) during two sessions, with a maximum of two weeks apart. The study was approved by the Ethical Committee of Leiden University Medical Center, the Netherlands.

### 2.2. Materials

#### 2.2.1. Schizotypal Symptoms

We used the Schizotypal Personality Questionnaire for Children, Dutch version (SPQ-C-D; [28]) to measure schizotypal symptoms in youth. This 74-item self-report questionnaire reflects subjective experience of schizotypal traits, and includes a total score and three dimensions: positive schizotypy, negative schizotypy, and disorganization [28]. All three dimension subscales have acceptable internal consistency (α’s > 0.81; [28]). In the current study, we used the total score (correlations between total score and dimensions > 0.73). Higher scores represent greater schizotypal symptoms.

#### 2.2.2. Social Anxiety

The Social Anxiety Scale (SAS; [29]) was employed to assess social anxiety. This questionnaire is suitable for children aged 8 and older. It consists of 36 items, each followed by two options: one option indicating social anxiety the other indicating no social anxiety. For example: ‘If someone in the group looks at me when I am doing something (1) I do not become nervous, (2) I become nervous’. Higher scores represent more social anxiety. In the current study, we used the total score, including social anxiety in four types of situation in which certain skills are at stake: social skills, intellectual skills, physical skills and appearance. The internal consistency of the total score is high, Cronbach’s alpha = 0.90 [29].

#### 2.2.3. Cognitive Coping Strategies

Cognitive coping was measured with the Cognitive Emotion Regulation Questionnaire - kids (CERQ-k; [18]), which assesses how children tend to think after experiencing negative life events. This 36-item questionnaire includes 9 subscales, each containing 4 items; for example, in the subscale Catastrophizing ‘I often think about how horrible the situation was’. Items are answered on a 5-point Likert scale ranging from *(almost) never* to *(almost) always*. The three subscales used in the current study have shown acceptable reliability in previous studies with samples with similar ages [15,18].

#### 2.2.4. Intellectual Functioning

IQ was assessed using the subtests Block design and Vocabulary of the Dutch adaptation of the Wechsler Intelligence Scales for Children [30]. This is called the V-BD short form and it is used to estimate full-scale IQ (FSIQ; [27]). Scores on the V-BD correlate highly with FSIQ (*r* = 0.88; [27]), and it is a valid estimate of intelligence with good reliability and validity [31].

### 2.3. Statistical Analyses

Data were analyzed using IBM SPSS, version 23.0. Pearson’s correlations were computed to investigate associations among schizotypal symptoms, social anxiety symptoms and cognitive coping strategies.

To investigate mediating mechanisms, a moderated mediation analysis was conducted using the PROCESS macro from Hayes [32,33], model 58. In this model, social anxiety (total score) was entered as the outcome variable, schizotypal symptoms (total) as independent variable, and the three coping strategies as parallel mediators. Group was included as the moderator on the dependent variables coping strategy and social anxiety. Gender and IQ were included as covariates in this model. See Figure 1 for a representation of this model.

The PROCESS macro was retrieved from www.processmacro.org. The macro uses ordinary least squares (OLS) analysis for calculating the mediation and moderated mediation effects, and bootstrapping for calculating the confidence intervals (CI). We used bias-corrected bootstrap CIs based on 5000 bootstrap samples with a 95% level of confidence. When the confidence intervals do not include zero, the effect is interpreted as significant. Bootstrapping “makes no assumptions about the shape of the distributions of the variables or the sampling distribution of the statistic” [34] (p. 722).

## 3. Results

### 3.1. Correlations among Schizotypal Symptoms, Social Anxiety Symptoms and Coping Strategies

Table 2 shows the Pearson correlations among study variables. Schizotypal symptoms correlated positively with social anxiety in both the extra X chromosome group (*r* = 0.76) and the control group (*r* = 0.51). The correlation in the extra X chromosome group is significantly larger than the correlation in the normal control group, Fisher’s *r*-to-*z* transformation = 2.14, *p* < 0.05 (two-tailed).

As regards the correlations between measures of psychopathology, schizotypal and social anxiety, with the three cognitive coping strategies, somewhat similar patterns emerged in the extra X chromosome group and the control group. Schizotypal symptoms correlated positively with all three strategies in both groups. These correlations are small to medium in strength, with only the association between schizotypal symptoms and blaming others in the extra X chromosome group not reaching significance. In terms of the correlations with social anxiety, only the relationship with Catastrophizing was significant in the extra X chromosome group. In both groups, correlations among the three coping strategies are small to moderate in size, and all except one significant. In the control group, Rumination correlates positively with Blaming others at trend level.

### 3.2. Moderated Mediation Model

The moderated mediation model 58 showed that the indirect effect was significant only for Catastrophizing and that the Catastrophizing × Group interaction was significant (path *b*), not path *a*. We therefore followed this model up with a simpler model (model 14), in which Catastrophizing was included as the only mediating variable, and the moderating effect of group on path *b* was tested. These results are presented in Table 3.

Schizotypal symptoms significantly positively predicted Catastrophizing (path *a*), coefficient = 0.05 (95% CI: 0.02 to 0.07). As regards the prediction of social anxiety, the direct effect (path *c’*) of schizotypal symptoms was significant, coefficient = 0.21 (95% CI: 0.16 to 0.27). The interaction between Catastrophizing and Group (path *b*) was significant, coefficient = 0.92 (95% CI: 0.26 to 1.58). The indirect effect of schizotypal symptoms predicting social anxiety through Catastrophizing was conditional on Group. The indirect effect was significant in the extra X chromosome group, coefficient = 0.04 (95% CI: 0.01 to 0.09) and nonsignificant in the control group, coefficient = −0.01 (95% CI: −0.03 to 0.01). This was confirmed by the index of moderated mediation, index = 0.04 (95% CI: 0.01 to 0.10). Hence, Catastrophizing mediates the association between schizotypal symptoms and social anxiety for the extra X chromosome group but not for the control group.

Post hoc moderated mediation analyses were performed to check whether the indirect effect applied to all three subscales of schizotypal symptoms, positive, negative and disorganization. These analyses (model 14), showed that the indirect effect of schizotypal symptoms on social anxiety through catastrophization was conditional on group and significant in the extra X chromosome group for all three dimensions (indirect effect positive symptoms = 0.09, 95% CI: 0.02 to 0.23; indirect effect negative symptoms = 0.08, 95% CI: 0.01 to 0.20; indirect effect disorganization symptoms = 0.14, 95% CI: 0.03 to 0.30).

## 4. Discussion

The aims of our study were to investigate the relationships between schizotypal symptoms, social anxiety, and cognitive coping styles, and to examine whether these cognitive coping styles mediated the relation between schizotypal symptoms and social anxiety in a group of youth with an extra X chromosome and a comparison group of typically developing peers. Based on previous research, three coping styles were selected as potential mediators: rumination, catastrophizing and blaming others. We found moderate-to-strong positive associations between schizotypal symptoms, social anxiety and rumination, catastrophizing, and blaming others in the extra X chromosome group. The correlations in the group of typically developing peers were generally smaller. Catastrophizing, not rumination or blaming others, was found to mediate the relationship between schizotypal symptoms and social anxiety, but only in youth with an extra X chromosome.

In both groups, schizotypal symptoms were positively associated with social anxiety, indicating that youth reporting a higher subjective experience of schizotypal symptoms report feeling nervous in different social situations. This relationship was significantly larger in the group with an extra X chromosome than in the group of typically developing youth. Extending previous work independently showing higher mean levels of both schizotypal and social anxiety symptoms [11,14] in youth with an extra X chromosome, the strong association between these symptoms is in line with the suggestion that this group is at-risk for the development of psychopathology [13].

The correlations between schizotypal and social anxiety symptoms on the one hand and the coping strategies on the other revealed somewhat similar patterns in the two groups. For typically developing youth and those with an extra X chromosome, higher schizotypal symptoms were associated with more frequent use of the coping strategies rumination, catastrophizing and blaming others. However, in terms of social anxiety and coping strategies, none of the correlations were significant in the typically developing group, with the correlation between social anxiety and catastrophizing near zero. In contrast, catastrophizing was significantly positively associated with social anxiety in the extra X chromosome group. The correlation with rumination was marginally significant (*p* = 0.06) suggesting that it would reach significance in a larger sample.

In terms of the extra X chromosome group these findings are in line with previous research showing that maladaptive coping, conceptually similar to rumination (amongst other strategies), is positively associated with schizotypal symptoms in youth at-risk for developing psychosis [20,22]. Moreover, we extend previous findings by showing a positive relationship between schizotypal symptoms and the coping strategies catastrophizing and blaming others (the latter not significant, possibly explained by the sample size). Indeed, the correlation with catastrophizing was equal in size to that with rumination, suggesting that youth with an extra X chromosome reporting schizotypal symptoms are likely to frequently ruminate and catastrophize in response to stressful life events. Moreover, our findings also indicate that in youth with an extra X chromosome reporting high social anxiety levels, catastrophizing is a frequently used coping strategy. To the best of our knowledge, this is the first study to report on associations between social anxiety and cognitive coping strategies in youth with an extra X chromosome, hence further studies are required to replicate these findings.

In typically developing youth, higher self-reported schizotypal symptoms but not social anxiety symptoms were associated with more frequent use of rumination, catastrophizing and blaming others. These positive and significant associations contrast with the results reported by Jalbrzikowski et al. [22]. In their study, schizotypal symptoms as measured by clinicians using the Structured Interview for Prodromal Symptoms [35] were not significantly correlated with maladaptive coping strategies as reported by the control group of participants. The different findings could be explained by how coping is measured in each study. The maladaptive coping subscale used by Jalbrzikowski et al. [22] contains items measuring behavioral and cognitive styles, whereas the CERQ used in the current study is a conceptually purer measure of cognitive coping styles [19]. Indeed, their maladaptive coping subscale did not include items that measure catastrophizing as a form of coping. Alternatively, the different findings could be explained by the fact that Jalbrzikowski et al. [22] correlated clinician-report with self-report, whereas our correlations are based on self-report only. Given that correlations from the same source are generally larger than between different sources, it is important that the current associations between schizotypal symptoms and cognitive coping strategies are replicated in a typically developing youth sample.

The non-significant correlations between social anxiety and cognitive coping strategies in our healthy control group are not in line with previous research where significant positive associations between anxiety, rumination, catastrophizing and blaming others [15] or between rumination and social anxiety [24] were reported. These inconsistent results could be attributed to differences in the measurement of internalizing symptoms and coping strategies. In the Garnefski and Kraaij [15] study, generalized anxiety symptoms were investigated in relation to cognitive coping strategies, not anxiety in relation to social situations specifically. In the Miers et al. [24] study, rumination was measured through a post-event processing questionnaire, developed to measure a key cognitive construct of social anxiety disorder and as described in key theoretical models [36,37]. Hence, in the latter study, the rumination items were likely conceptually more aligned to social anxiety symptoms than the rumination items as measured by the CERQ used in the current study. Further studies investigating the relationships between cognitive coping strategies and social anxiety specifically in typically developing youth are therefore recommended.

Of the three coping strategies, catastrophizing was found to be a significant mediator of the relationship between schizotypal symptoms and social anxiety in the extra X chromosome group only. In this group, the path from schizotypal symptoms to catastrophizing was positive and significant as was the path from catastrophizing to social anxiety. Schizotypal symptoms predicted social anxiety both directly and indirectly through catastrophizing coping. Moreover, the mediated indirect link held for all three schizotypal dimensions: positive, negative and disorganized. These findings held after controlling for gender and IQ. In typically developing youth, the path from schizotypal symptoms to catastrophizing was positive and significant, as was the direct effect between schizotypal and social anxiety symptoms. However, the path from catastrophizing to social anxiety was non-significant, and catastrophizing coping did not mediate the schizotypal–social anxiety link.

To our knowledge, this is the first study to show that using catastrophizing as a coping strategy to threatening events is a potential risk factor for social anxiety (disorder) in a population at-risk for clinical levels of schizotypal symptoms, that is, youth with an extra X chromosome. Our moderated mediation model showed that, in youth with an extra X chromosome, those who experience schizotypal symptoms are likely to have high social anxiety levels and that this link is partly explained by a tendency to catastrophize in response to threatening events. Hence, a greater occurrence of catastrophizing thoughts, such as thinking how horrible the situation is and how it is the worst thing that can happen, increase feelings of nervousness in social situations where one’s social, intellectual and physical skills and appearance are at stake. These findings suggest that in youth with an extra X chromosome and elevated self-reported schizotypal symptoms, attention should be given to their coping strategies in order to reduce the risk of maintaining or developing social anxiety symptoms. More specifically, our results attest to the relevance of catastrophizing specifically, versus other “maladaptive” forms of coping such as rumination and blaming others.

Limitations of this study are differences in the level of intellectual functioning, sample size, use of self-report measures, and cross-sectional design. First, even after selection of participants with an IQ greater than or equal to 70, the extra X chromosome group had a significantly lower mean IQ level as compared to the control group. However, we controlled for IQ in the mediation analyses, hence the mediation effect cannot be explained by differences in IQ. Second, the extra X chromosome group was relatively small as compared to the comparison group, which could have affected the study’s power to detect mediation effects by all three coping strategies. It is plausible that rumination too might mediate the schizotypal–social anxiety link in a larger sample of youth with an extra X chromosome, given its marginal significant correlation with social anxiety and significant correlation with schizotypal symptoms. Studies with a larger sample size are required to investigate this possibility. Third, the current study’s results are derived from self-report measures, increasing the chance of finding significant associations between independent and dependent variables. Nevertheless, given the subjective (e.g., psychological well-being and cognitive coping strategies) nature of the constructs assessed, self-report is arguably a valid method. Moreover, we selected participants aged 8 or above as recommended for reliable assessment of these constructs [18]. At the same time, using different informants and/or more objective measurements, for example, clinical assessment of schizotypal and social anxiety symptoms could bolster the current findings. Finally, the cross-sectional design means that we cannot assert a temporal relationship between schizotypal symptoms, catastrophizing coping, and social anxiety. It is equally plausible that social anxiety symptoms precede schizotypal symptoms, possibly mediated by cognitive coping strategies. Indeed, several studies do point to social anxiety as a potential precursor to schizotypal symptom development [38,39]. Further research employing a longitudinal design in which psychopathological symptoms and coping strategies are assessed over time in youth with an extra X chromosome and compared to a group of typically developing peers is required to tease apart causality effects and identify specific risk factors for the development of schizotypal symptoms versus social anxiety in these groups.

## 5. Conclusions

In conclusion, the current study’s findings implicate cognitive coping as an important psychological concept to assess in individuals susceptible to psychopathology, children and adolescents with an extra X chromosome. In particular, catastrophizing seems to be a key marker of risk, especially when increased schizotypal and social anxiety levels are evident. Youth with Klinefelter or Trisomy X syndrome who experience a loss of control and grip on reality, one’s own emotions, thoughts and behaviors, may benefit from interventions targeted to this coping style. Focusing on coping with stressors in the environment may help to prevent the development of severe psychotic symptoms and social anxiety in this vulnerable group [20]. The main finding that catastrophizing mediates the schizotypal–social anxiety relationship may contribute to the understanding of how social anxiety (disorder) develops in children and adolescents with the Klinefelter or Trisomy X syndrome.

## Figures and Tables

**Figure 1 brainsci-07-00113-f001:**
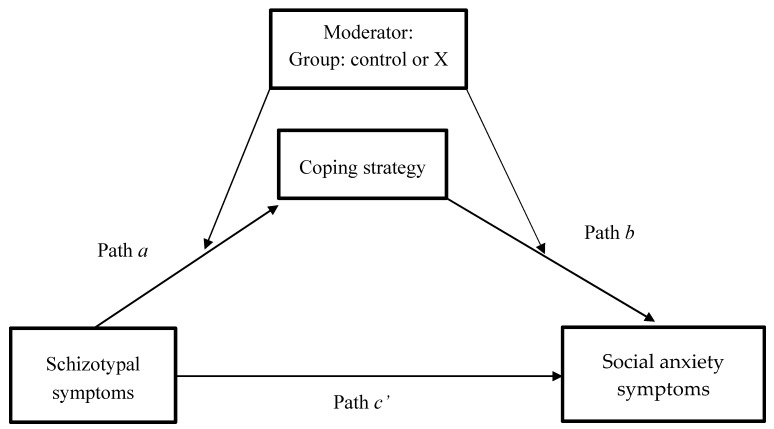
Diagram of the moderated mediation model with coping strategy as the mediator and group as the moderator (model 58).

**Table 1 brainsci-07-00113-t001:** Characteristics of the extra X chromosome and the control groups (scores represent means and standard deviations).

Variable	Control Group (*n* = 107)	XXY or XXX Group (*n* = 34)	Group Difference
Gender B/G	47/62	22/16	χ^2^ (1) = 2.47, ns
Age (years)	12.03 (3.02)	12.97 (3.09)	F (1, 139) = 2.51, ns
IQ	103.27 (13.36)	88.84 (11.53)	F (1, 139) = 32.07, *p* < 0.001
Parent education level ^a^	2.14 (0.64)	2.31 (0.63)	F (1, 139) = 1.72, ns

^a^ Ranges from 0 (primary school) to 3 (university).

**Table 2 brainsci-07-00113-t002:** Pearson Correlations between Schizotypal Symptoms, Social Anxiety Symptoms, and Cognitive Coping Strategies.

Variable	1	2	3	4	5
1. Schizotypal symptoms	-	*0.76 ***	*0.42 **	*0.43 **	*0.20*
2. Social anxiety	0.51 **	-	*0.33*	*0.52 ***	*0.17*
3. Rumination	0.28 **	0.16	-	*0.51 ***	*0.40 **
4. Catastrophizing	0.29 **	0.09	0.55 **	-	*0.64 ***
5. Blaming others	0.35 **	0.16	0.19 ^a^	0.20 *	-

*Note* Correlations for the extra X chromosome group presented in the top diagonal and for the control group in the bottom diagonal. *N*’s in the extra X chromosome group range between 34 and 36; *N*’s in the control group range between 108 and 109. ** *p* < 0.01; * *p* < 0.05. ^a^
*p* = 0.05.

**Table 3 brainsci-07-00113-t003:** Moderated Mediation Results for the Link between Schizotypal Symptoms and Social Anxiety with 95% Bias-corrected Confidence Intervals (*n* = 142).

Moderated Mediation Results	Coefficient	LLCI	ULCI
Outcome: Catastrophizing			
*R* = 0.28, *F* (3, 138) = 4.05, *p* < 0.05			
Schizotypal symptoms	0.05	0.02	0.07
Gender	0.23	−0.81	1.27
IQ	−0.01	−0.05	0.03
Outcome: Social anxiety			
*R* = 0.70, F (6, 135) = 21.87, *p* < 0.001			
Catastrophizing	−0.13	−0.44	0.18
Schizotypal symptoms	0.21	0.16	0.27
Group	−5.13	−10.59	0.33
Catastrophizing x Group	0.92	0.26	1.58
Gender	−0.64	−2.37	1.10
IQ	−0.02	−0.08	0.05
Conditional indirect effect at	Effect	LLCI	ULCI
Control group	−0.01	−0.03	0.01
Extra X chromosome group	0.04	0.01	0.09
Direct effect	0.21	0.16	0.27

*Note*. LLCI: Lower Limit Confidence Interval; ULCI: Upper Limit Confidence Interval.

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
