# Peer review of "Connecting the Dots between Schizotypal Symptoms and Social Anxiety in Youth with an Extra X Chromosome: A Mediating Role for Catastrophizing"

_brainsci, 2017, doi:10.3390/brainsci7090113_

Round 1

Reviewer 1 Report

It is known that individuals with an extra X-chromosome display higher levels of schizotypal symptoms and social anxiety as compared to "normally" developing youth. Using a battery of tests the authors of the  current study investigated how schizotypal and social anxiety symptoms are related. From their results the authors conclude  "that youth with an extra X-chromosome with schizotypal symptoms could benefit from intervention to weaken the tendency to catastrophize as a way of reducing the likelihood of social anxiety symptoms".

Concerns: Since the journal "Brain Sciences" is mainly read by basic and clinical neuroscientists, it would be ingenious to add some ideas of how an extra X-chromosome might influence cognition and anxiety-related behavior. What is known about this extra X? Is it (partly )silenced/inactivated? What about lngRNAs in this context? Does imprinting phenomena playi a role? Are there any data showing a specific Impact of extra X on brain biology?

Author Response

Response: We have added a paragraph (lines 44-55) that contains hypotheses about the role of the extra X-chromosome in the Klinefelter and Trisomy X phenotypes, specific gene overexpression, and how the altered brain development may be related to cognition, behavior and psychopathology. 

Reviewer 2 Report

The higher rates of schizotypal symptoms and social anxiety among individuals with an extra X-chromosome have been already reported in literature. However, in this study the authors performed a wider analysis, investigating simultaneously social anxiety, schizotypal symptoms, intellectual functioning and cognitive styles. The latter in particular have been analyzed as possible mediators of the association between social anxiety and schizotypal symptoms. Improving the knowledge on the impact of cognitive styles on the patients' psychopathological manifestations, also considering that schizotypal symptoms can be imagined along a continnum from normality to schizophrenia, is fundamental to identify future therapeutic strategies.

The study design is good, the statistycal analysis accurate.

I would suggest to maintain the data reported in the introduction, but shortening the lenght a little. Having done so, the authors could report more data on the genetics of behavior and cognition. In fact, they refer to the "...exceptional high density of genes on the X‐chromosome that are essential for neural, and thus cognitive and behavioral development" without further information.

PS: in line 37, there is a typo: having AND (instead of an) extra...

Author Response

Responses: 

We have trimmed the Introduction, particularly the discussion of evidence relating to coping styles and psychopathology. 

We have added a paragraph (lines 44-55) that contains hypotheses about the role of the extra X-chromosome in the Klinefelter and Trisomy X phenotypes, specific gene overexpression, and how the altered brain development may be related to cognition, behavior and psychopathology. 

We have checked the text and do not agree that it is a typo. The text reads ‘individuals with an extra X-chromosome’. If we add a ‘d’ it will incorrectly read ‘with and extra X-chromosome’.